# Research on an Improved Non-Destructive Detection Method for the Soluble Solids Content in Bunch-Harvested Grapes Based on Deep Learning and Hyperspectral Imaging

**Junhong Zhao** [1,2], **Qixiao Hu** [3], **Bin Li** [1,2,*], **Yuming Xie** [1,2], **Huazhong Lu** [2,4] and **Sai Xu** [1,2]

1. Institute of Agriculture, Guangdong Academy of Agricultural Sciences, Guangzhou 510640, China; zhaojunhong@gdaas.cn (J.Z.); xieyuming@gdaas.cn (Y.X.); xusai@gdaas.cn (S.X.)
2. Guangdong Laboratory for Lingnan Modern Agriculture, Guangzhou 510640, China; huazlu@scau.edu.cn
3. College of Engineering, South China Agricultural University, Guangzhou 510642, China; huqixiao@163.com
4. Guangdong Academy of Agricultural Sciences, Guangzhou 510640, China
* Correspondence: libin@gdaas.cn; Tel.: 86-180-1199-8740

**Featured Application: Non-destructive detection and quality grading of the internal quality of grapes after harvest.**

**Abstract:** The soluble solids content (SSC) is one of the important evaluation indicators for the internal quality of fresh grapes. However, the current non-destructive detection method based on hyperspectral imaging (HSI) relies on manual operation and is relatively cumbersome, making it difficult to achieve automatic detection in batches. Therefore, in this study, we aimed to conduct research on an improved non-destructive detection method for the SSC of bunch-harvested grapes. This study took the Shine-Muscat grape as the research object. Using Mask R-CNN to establish a grape image segmentation model based on deep learning (DL) applied to near-infrared hyperspectral images (400~1000 nm), 35 characteristic wavelengths were selected using Monte Carlo Uninformative Variable Elimination (MCUVE) to establish a prediction model for SSC. Based on the two above-mentioned models, the improved non-destructive detection method for the SSC of bunch-harvested grapes was validated. The comprehensive evaluation index $F_1$ of the image segmentation model was 95.34%. The $R_m^2$ and RMSEM of the SSC prediction model were 0.8705 and 0.5696 Brix%, respectively, while the $R_p^2$ and RMSEP were 0.8755 and 0.9177 Brix%, respectively. The non-destructive detection speed of the improved method was 16.6 times that of the existing method. These results prove that the improved non-destructive detection method for the SSC of bunch-harvested grapes based on DL and HSI is feasible and efficient.

**Keywords:** grape; bunch-harvested; hyperspectral imaging; deep learning; non-destructive detection

## 1. Introduction

The grape is one of the four major fruits in the world. It has a bright color and delicious taste and is well-liked by consumers. It is an important high-value-added crop in China. Moreover, China is the largest producer and consumer of table grapes [1,2]. With the increase in the population's consumption level, the demand for high-quality grapes is growing rapidly. Table grapes are mainly harvested and sold in whole bunches. The realization of the rapid, non-destructive detection of the internal quality of bunch-harvested grapes is of great significance for guiding production, improving grape sale profits, and improving the grape grading system [3].

The soluble solids content (SSC) is an important indicator for judging the quality and ripeness of grapes, and it is a determinant of the overall taste and flavor of grapes [4]. Since the SSC of grapes cannot be estimated manually by the naked eye, destructive sampling is often used to assess the ripeness of the entire batch. This method not only requires the

destruction of grape berries, causing waste in production, but also has a low efficiency and cannot accurately evaluate the quality of each bunch of grapes; thus, it is difficult to achieve the accurate grading of large quantities in production.

In recent years, HSI has gradually been applied for the non-destructive detection of fruits [5], such as tomatoes [6], loquats [7], blueberries [8], red jujubes [9], apples [10], etc. Hyperspectral data can be represented by a "3D image cube", which can simultaneously obtain image and spectral information on fruits, as shown in Figure 1. The horizontal and vertical coordinate information of the two-dimensional image is represented by $x$ and $y$, respectively. The third dimensional spectral information is represented by $\lambda$. The detection technology based on HSI uses the reflectance characteristics of the target object in regard to different spectra to predict the internal characteristics of the target. Related studies on the non-destructive detection of the internal quality of grapes have been conducted.

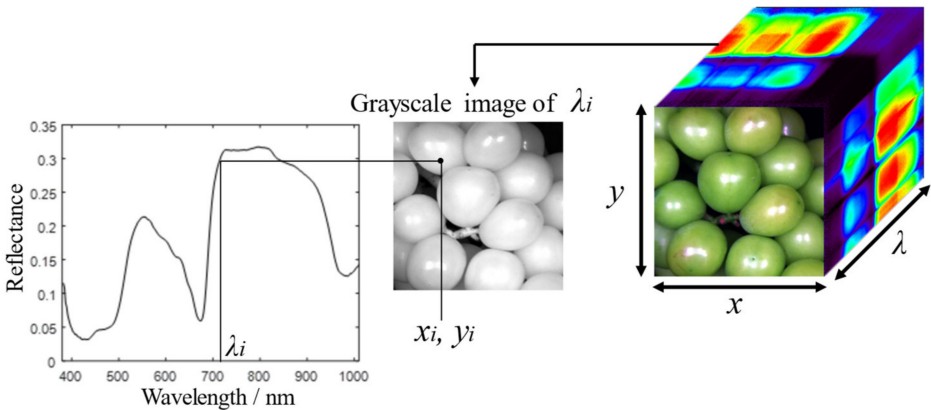

**Figure 1.** Schematic diagram of hyperspectral 3D image cube.

In current research on hyperspectral imaging, line-scan spectroscopy (pushbroom) and point spectroscopy (whiskbroom) are mainly used. In these studies, relatively accurate prediction models for SSC have been established. In research using line-scan spectroscopy [11–14], the reflectance spectrum of multiple grapes can be obtained in one image, but the methods require the manual selection of the Region of Interest (RoI) to obtain the average spectral reflectance, which cannot be automatically detected in batches. In research using point spectroscopy [15,16], the reflectance spectrum of one grape can be obtained each time; still, this one-by-one detection method has a low efficiency. Therefore, the applicability of current research is limited: for example, non-destructive detection of a small amount of grape berries in the laboratory. To achieve the non-destructive detection of every single grape berry in order to accurately grade the quality of grape bunches, a large amount of time would be required.

Aiming to address the defects of the existing non-destructive detection technology based on HSI, in this article we propose an improved method. By combining the existing method with target recognition and image segmentation technology based on DL, we extracted the texture feature information in the image to establish an accurate model to identify and segment grapes [17]. In this way, the RoI can be extracted automatically and efficiently so as to obtain the average spectral reflectance of each grape.

In this study, the existing non-destructive detection method for grape SSC based on near-infrared hyperspectral (400~1000 nm) images was improved. Taking Shine-Muscat, whose maturity is difficult to distinguish with the naked eye, as the research object, combined with the image segmentation algorithm Mask R-CNN based on DL, an image segmentation model and an SSC prediction model were established to improve the efficiency of the non-destructive detection of the SSC in whole bunches of grapes.

## 2. Materials and Methods

Research on the non-destructive detection method of the SSC in bunched grapes led to the establishment of two detection models, namely the SSC prediction model based on HSI and the grape instance segmentation model based on DL.

The non-destructive testing process for SSC in bunch-harvested grapes based on the above two models is shown in Figure 2. The steps are as follows: obtain the hyperspectral images of the whole bunch of grapes through the hyperspectral image acquisition platform; obtain the pixels of each single grape through the Mask R-CNN instance segmentation algorithm; obtain the spectral data of each grape in the image and establish the dataset of the bunch of grapes; predict the SSC of each grape through the SSC prediction model; and obtain the SSC of the bunch of grapes.

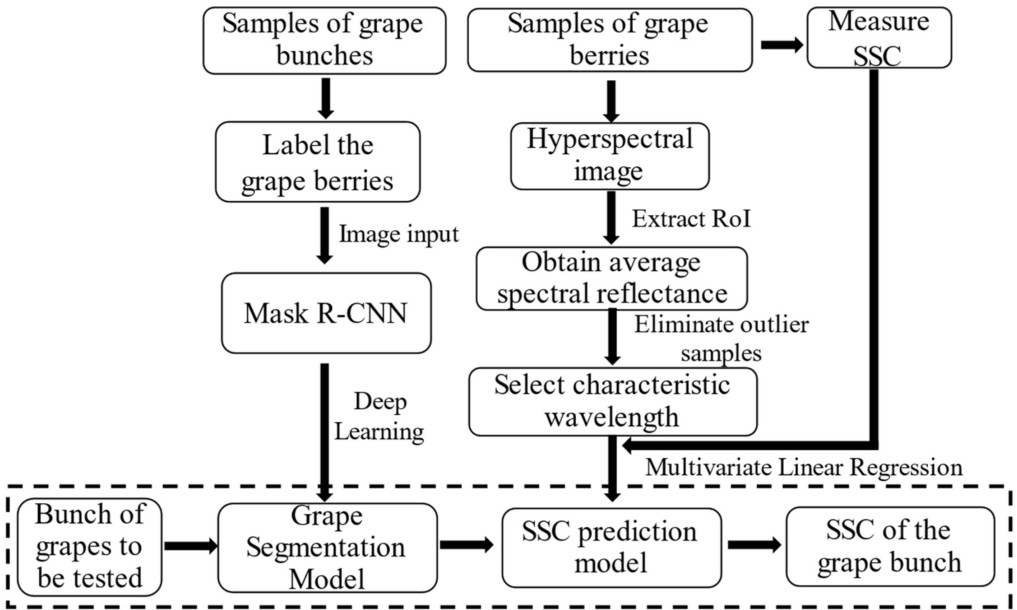

**Figure 2.** Flowchart of research object and methods.

### 2.1. Objects and Devices

The grape variety selected for this experiment was Shine-Muscat, with a total of 23 bunches of grapes and a total of 345 grape samples. On 9 July, 22 July, 12 August, 30 December 2021, and 11 January 2022, the grapes were picked in the greenhouse of Baiyun Base of Guangdong Academy of Agricultural Sciences. The picked bunches had varying degrees of ripeness, smooth surfaces, and defect-free grapes. Based on the experience of cultivators, 5 bunches were around 13 Brix%, 6 bunches were around 16 Brix%, 7 bunches were around 19 Brix%, and 5 bunches were around 22 Brix%. The relevant testing equipment was as follows:

1.  Portable line-scan hyperspectral imager GaiaField-V10 (Sichuan Dualix Spectral Imaging Technology Co., Ltd., Chengdu, China) with built-in pushbroom (as shown in Figure 3). Response spectrum: 400~1000 nm; maximum resolution: 1392 × 1040; number of spectral channels: 256; spectral resolution: 3.5 nm; protocol: USB 2.0. The dedicated light source was HSIA-LS-T-200W, consisting of four halogen lamps. The standard whiteboard was HSIA-CT-150 mm × 150 mm, made of PTFE, with nominal reflectance of 99%.

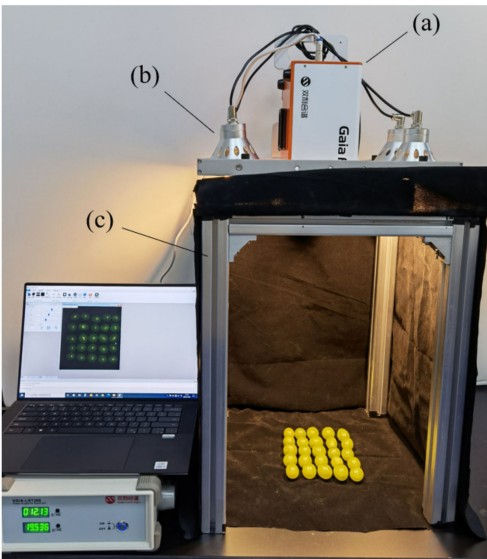

**Figure 3.** Hyperspectral image acquisition platform. (**a**) Hyperspectral imager. (**b**) Halogen lamps. (**c**) Camera obscura.

2.  Digital refractometer PAL-1 (ATAGO Co., Ltd., Tokyo, Japan): measuring range: Brix0.0~53.0%; measuring accuracy: Brix $\pm$ 0.2%; measuring temperature: 10~100 °C; working environment temperature: 10~40 °C, used to measure the soluble solid content of grapes.
3.  Tower server Dell T640 (Dell Inc., Round Rock, USA): OS: Ubuntu 18.04; CPU: Intel Xeon(R) Gold 5218 @ 2.30 GHz; GPU: Nvidia GeForce RTX 3090 $\times$ 2; RAM 64 GB; ROM 4 TB), used for establishing the grape segmentation model.

### 2.2. Method of Establishing the Grape Segmentation Model

In practical application scenarios, there are differences in the background, the distribution of grape berries is relatively dense, and there are problems such as overlapping and shadowing, resulting in unclear edges of the grape berries, which affects the rapid identification of individual grapes in grape clusters. Traditional image processing algorithms based on edge feature extraction are not accurate enough to meet the requirements.

The instance segmentation algorithm Mask R-CNN, proposed by Kaiming He et al. in 2017, can effectively identify the target while completing high-quality instance segmentation. Instance segmentation can be seen as a combination of target detection and semantic segmentation. Relative to the bounding box of target detection, instance segmentation can be accurate up to the edge of the object; relative to semantic segmentation, instance segmentation can distinguish different instances on the basis of specific categories [18]. In short, it can not only accurately segment the classification of "grape berry", but also distinguish different grape berries. The network not only has a high detection accuracy but also runs at 5 frames per second under a single GPU, thus basically meeting the requirements for real-time instance segmentation. Therefore, considering all aspects, we used the image instance segmentation algorithm Mask R-CNN [19] based on deep learning.

### 2.2.1. Image Acquisition and Annotation

1.  Obtain the training set and validation set. The hyperspectral imager could not directly capture RGB images, but a total of approximately 50 images were collected in both laboratory and greenhouse with pseudo-RGB images (R-638.8 nm, G-550.2 nm, B-459.4 nm) synthesized from hyperspectral images of three wavelengths (as shown in Figure 4). The split ratio of the training set and validation set was 4:1 [20].

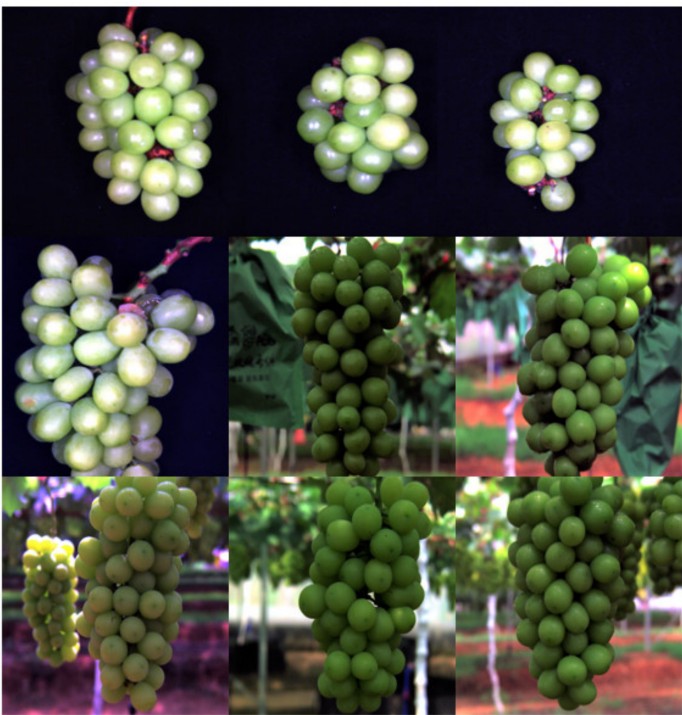

**Figure 4.** Part of the image dataset used for deep learning.

2. Annotate the image. Since it was necessary to train a model that could accurately identify grapes with smooth surfaces and no defects, grapes with no damage on the surface and no more than 50% occlusion of the area were selected for stroke in the training set. In this experiment, the labeling tool LabelMe was used for labeling [21].

### 2.2.2. Training with Mask R-CNN

In this experiment, we used the Detectron2 algorithm library to implement the Mask R-CNN instance segmentation algorithm. Detectron2 is a second-generation CV library launched by Facebook AI Research which integrates algorithms such as target detection, instance segmentation, pose estimation, semantic segmentation, and panoramic segmentation and has been implemented in recent years [22].

Mask R-CNN has the following main parts: image input, a Convolutional Neural Network backbone composed of ResNet and Feature Pyramid Network (FPN), a feature map, Region Proposal Network (RPN), RoIAlign, and image output (mask, class, bounding box) [19,23]. The network map is shown in Figure 5.

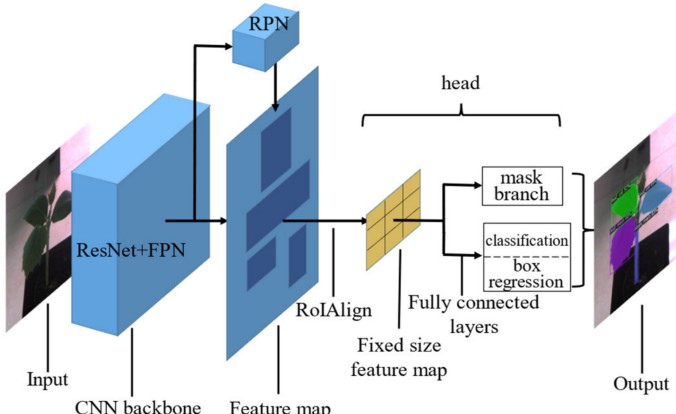

**Figure 5.** Mask R-CNN network map.

The process was as follows. First, we used the default parameters for training. According to the segmentation effect of the trained model, we determined whether the segmented grapes were complete, whether the edges were accurate, etc. Then, we adjusted the parameters for training and repeatedly optimized them to obtain the optimal parameters.

### 2.2.3. Model Evaluation

After the training of the model was completed, the identification results of the grapes needed to be evaluated. We used the $F_1$ score to evaluate the model's performance. It is defined as the harmonic mean of the precision rate and recall rate. The definition of $F_1$ is shown in Equations (1)–(3):

$$P = TP \ / \ (TP + FP) \tag{1}$$

$$R = TP \ / \ (TP + FN) \tag{2}$$

$$F_1 = 2PR \ / \ (P + R) \tag{3}$$

In the equations, $P$ is the precision rate and $R$ is the recall rate. $TP$ is the number of grapes that are actually grapes and predicted by the model as grapes, that is, the actual positive samples that are detected as positive samples. $FP$ is the number of grapes that are actually background but predicted by the model as grapes, that is, the actual negative samples that are detected as positive samples. $FN$ is the number of grapes that are actually grapes but not identified as grapes, that is, the actual positive samples that are not detected as positive samples. The closer $F_1$ is to 1, the better the performance of the model is [24].

### 2.3. Method of Establishing the SSC Prediction Model

The SSC prediction model based on hyperspectral images is key to the detection accuracy. Its main function is to accurately estimate the SSC of a single grape based on its spectral reflectance characteristics. The main process of the research using this model is as follows:

1.  Establish the input set: collect images in the 400 nm–1000 nm spectral band of grapes of different levels of maturity through the hyperspectral acquisition platform; perform black and white correction on the hyperspectral images to obtain the standard reflectance.
2.  Establish the output set: detect the SSC of each grape that has completed hyperspectral imaging; select the RoI to obtain the average spectral data of each grape; take the SSC as the label value of the average spectral data.
3.  Establish the modeling set: based on the input set and the output set, a modeling set and a test set are obtained for the establishment and evaluation of the SSC prediction model.
4.  Establish the SSC prediction model: adopt different methods to select characteristic wavelengths and establish different models; evaluate different models using the test set to select the best model.

### 2.3.1. Establishment of SSC Prediction Model Sample Set

(1) Hyperspectral image acquisition and correction

1.  The acquisition of modeling set samples was as follows. Five grapes were picked randomly and equidistantly from top to bottom in the same vertical direction from the same bunch of grapes (as shown in Figure 6a,b). We picked 5 grapes, working in both vertical directions, from the same bunch of grapes for a total of 10 grapes. A total of 100 grapes from 10 bunches (2 bunches of 20% ripeness, 3 bunches of 50% ripeness, 3 bunches of 70% ripeness, and 2 bunches of 100% ripeness) were collected and arranged in a 5 × 5 format in one image (Figure 6c). We used a hyperspectral

imager for imaging and turned on the light source, allowing it to warm up for half an hour before imaging to stabilize the current.

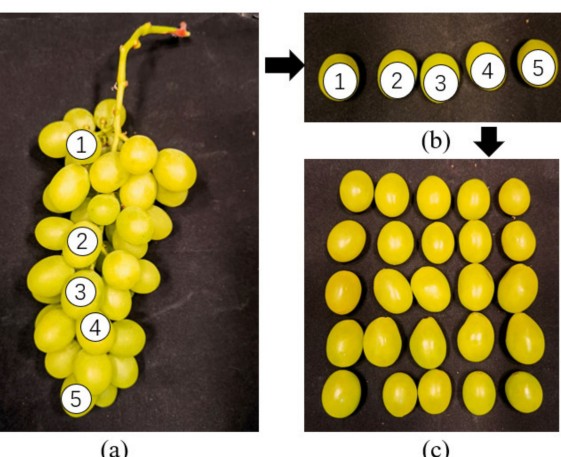

(a)      (c)

**Figure 6.** Hyperspectral image acquisition process (1–5 represent the label number of grapes). (**a**) A bunch of grapes. (**b**) Picked grape samples. (**c**) A set of samples for each hyperspectral image.

2.  The acquisition of test set samples was as follows. Five grapes were picked randomly and equidistantly from top to bottom in the same vertical direction from the same bunch of grapes (as shown in Figure 6a,b). We collected 5 grapes based on the same horizontal spacing, that is, $5 \times 5 = 25$ grapes were collected from the same bunch of grapes, forming one set of data. A total of 125 grapes were collected from 5 bunches, and they were placed in the same way as shown in Figure 6c above. We used the hyperspectral imager for imaging and turned on the light source, allowing it to warm up for half an hour before imaging to stabilize the current.

3.  The hyperspectral image correction was obtained as follows. After hyperspectral imaging, it was necessary to eliminate the influences of factors such as camera dark current and uneven illumination on the image and obtain a standard reflectance; thus, black and white correction was performed. The all-black calibration image $I_{dark}$ was obtained by covering the lens, the all-white image $I_{white}$ was obtained by scanning the standard whiteboard (nominal reflectance is 99%), and the corrected image was obtained according to formula (1) [25], where $I_{ref}$ is the calibrated image and $I_{raw}$ is the original hyperspectral image. The calibrated image contained 256 wavelengths of standard reflectance per pixel.

$$I_{ref} = (I_{raw} - I_{dark}) \ / \ (I_{white} - I_{dark}) \tag{4}$$

(2) Obtaining the average standard reflectance

The average spectral data of the grapes should correspond to the measured soluble solids content; hence, each grape berry must be numbered. For the sake of convenient numbering, in this experiment, we used ENVI 5.3, developed by Research System INC., to process the hyperspectral images of table grapes and extract data from the collected hyperspectral images. Four $20 \times 20$ square regions were selected, being equally spaced for each sample, using the polygon tool as the Region of Interest (RoI). When selecting regions, one should avoid overbright areas and dark-edged areas (as shown in Figure 7). The average spectral reflectance of all pixels in the RoI was taken as the spectral data of a single sample.

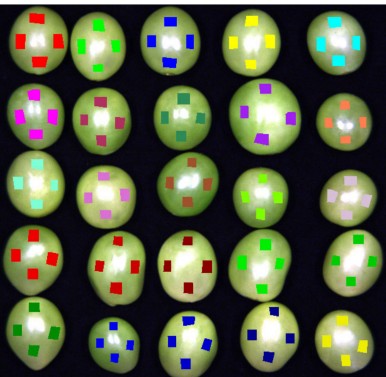

**Figure 7.** Selected RoI of hyperspectral images (different colors represent different RoIs).

(3) SSC measurement

After completing the collection of hyperspectral image samples, according to the Chinese national standard NY/T 2637-2014, "Refractometric method for determination of total soluble solids in fruits and vegetables", we squeezed out the grape juice and allowed the juice to drip onto the digital display refractometer mirror window. The data of the digital display refractometer were read and recorded. The measurement was repeated three times for each sample, and the average value was taken as the true value of a single sample [26].

(4) Eliminating outlier samples

Before selecting characteristic wavelengths and modeling, outlier samples should be eliminated to reduce the impact on the model's performance [27]. The Mahalanobis distance method is used to verify the distance between samples through spectral data and does not require SSC data, thus avoiding errors caused by human operation to a certain extent. Therefore, using the Mahalanobis distance method to eliminate outlier samples can help to improve the accuracy of the model [28].

In this research, the Mahalanobis distance method was used to eliminate outlier samples.

### 2.3.2. The Method of Establishing the SSC Prediction Model

(1) Selecting characteristic wavelengths

Since hyperspectral images have a reflectivity of 256 wavelengths, that is, data with 256 channels, not only is the amount of data large, but the data also contain redundant information and collinearity information. It has been proved that not all bands have equal importance [29]. It is necessary to delete redundant wavelength variables and select characteristic wavelengths that are highly correlated with the measured indicators so as to simplify and optimize the model and enhance the robustness of the model. The method of selecting the characteristic wavelength employs the Successive Projections Algorithm (SPA), the Monte Carlo Uninformative Variable Elimination (MCUVE), and the combination of the two methods.

- SPA is a forward variable selection method that obtains the subset of variables with minimal collinearity using the operation of projection in the vector space [30]. In this experiment, the number of spectral data samples $N$ and the number of wavelengths, 256, can form a spectral matrix $X_{N \times 256}$. SPA starts with one wavelength, calculates its projection on unselected wavelengths in each cycle, and then introduces the wavelength with the largest projection vector into the wavelength combination. The number of cycles is the number of selected characteristic wavelengths. Each newly selected wavelength has the smallest linear relationship with the previous wavelength. The algorithm models the results of each selection for prediction and analysis. If the predicted root mean square error (RMSE) of the modeling verification achieves the minimum value, the number of wavelengths selected at this time is the best [31].

- MCUVE combines the Monte Carlo method with the Uninformative Variable Elimination (UVE) method, aiming to address the large amount of redundant information in the spectral data while making full use of the intrinsic correlation between samples and evaluating the contribution of the wavelength in the spectral data to the measured index. Based on the contribution of each wavelength, wavelengths with no information are eliminated [32].

(2) Quantitative modeling and model evaluation

Partial Least Square Regression (PLSR) is a multivariate statistical data analysis method which is mainly used for the regression modeling of multiple dependent variables with respect to multiple independent variables [33]. In the modeling process, the PLSR method integrates the characteristics of Principal Component Analysis, Canonical Correlation Analysis, and Linear Regression Analysis methods; thus, a more reasonable multivariate linear regression model can be established [34].

After the regression model is established, the prediction effect of the model is measured using the following parameters: the main evaluation indicators are the coefficient of determination $R_p^2$ of the test set and the root mean square error RMSEP of the test set, while the auxiliary evaluation indicators are the $R_m^2$ of the modeling set and the RMSEM of the modeling set.

### 2.4. Validation of Improved Non-Destructive Detection Method for SSC

Eight bunches of grapes with different degrees of maturity were used as samples (as shown in Figure 8) to verify the non-destructive testing method for the SSC of bunched grapes (2 bunches of 20% ripeness, 2 bunches of 50% ripeness, 2 bunches of 70% ripeness, and 2 bunches of 100% ripeness).

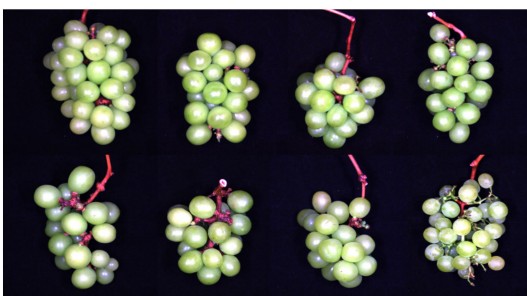

**Figure 8.** Validation samples of improved non-destructive detection method for SSC in bunch-harvested grapes.

After hyperspectral images of whole bunches of grapes were obtained, a digital refractometer was used to detect the SSC value of each grape in the image, and the value was determined in one-to-one correspondence with the position in the image.

#### 2.4.1. Grape Segmentation and Removal of Overbright Areas

Firstly, the Mask R-CNN instance segmentation model was used to identify and segment the grapes in the image and obtain the pixels corresponding to the grape berries whose occlusion area did not exceed 50% in a bunch.

Since the light source used for imaging in the platform came from four halogen lamps and the grape surfaces were smooth, the light was specularly reflected on the grape surface, causing some areas to be overbright visually, which greatly interfered with the prediction results. Therefore, it was necessary to remove the overbright areas. The grayscale image under a single wavelength was selected, binarized, and inverted as a mask for removing the overbright area and then multiplied with the original image to obtain grape pixels without overbright areas.

2.4.2. Grape SSC Predicted Value Visualization

Based on the spectral data of the pixels of each grape in the hyperspectral image, the SSC prediction model was used to obtain the SSC prediction value of each grape in the bunch, and it was displayed in the geometric center of each grape.

Finally, we compared the actual value with the predicted value to judge the validity and accuracy of the model.

2.4.3. Efficiency Comparison between the Existing Method and Improved Method

For the same batch of samples, the existing non-destructive detection method based on hyperspectral imaging and the improved method, respectively, were adopted. We recorded the time required for the two methods, respectively, and calculated the average SSC prediction time of each grape so as to determine the increase in the efficiency of the improved method.

## 3. Results and Discussions

### 3.1. Results and Analysis of the Grape Segmentation Test

3.1.1. Model Training

There are multiple official baseline models for Mask R-CNN on Detectron2 (https://github.com/facebookresearch/detectron2/blob/main/MODEL_ZOO.md, accessed on 9 August 2022), as shown in Table A1. It can be seen from the table that the benchmark model using the ResNet + FPN backbone network has a shorter training time and inference time and maintains a high box AP and mask AP, which means that it has the best speed/accuracy tradeoff. Since the complexity of the Shine-Muscat segmentation model was not high, ResNet-50 was selected as the feature extractor rather than ResNet-101. After comprehensive consideration, the benchmark model R50-FPN-3x was selected for training.

The main training parameters that affect the model's accuracy are the batch size and epoch. The batch size should not be too small or too large, because when the batch size is too small, there will not be sufficient time to converge during training. Under the common 100-epoch setting, the batch size is generally not less than 16. When the batch size is too large, not only will the memory of the GPU overflow, but there will also be problems with deep learning optimization and generalization [35].

After several parameter adjustments, the following training parameters were finally determined: the batch size was 16, the learning rate was 0.02, the batch size per image was 512, and the epoch number was 100.

3.1.2. Training Results

Applying the trained model to the validation set, it can be seen from the actual segmentation effect that the model could effectively identify the target, and the boundary segmentation was accurate, distinguishing the grape surface from the other parts (as shown in Figure 9).

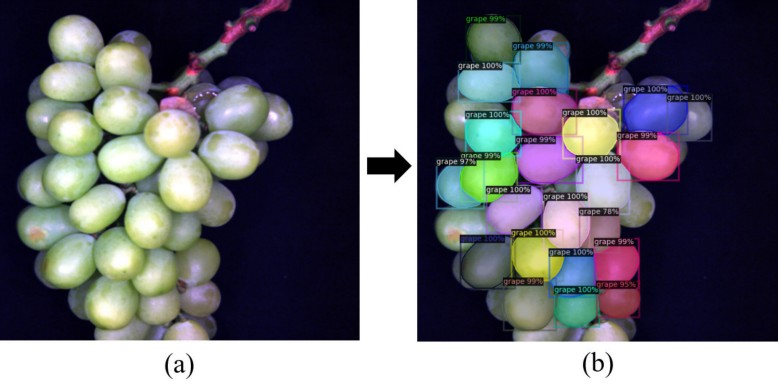

(a)                                    (b)

**Figure 9.** Instance segmentation effect. (**a**) Original image. (**b**) Segmentation result.

### 3.1.3. Model Evaluation

We randomly selected 20 grape images from the validation set and counted the number of detected positive samples (*TP*), the number of falsely detected positive samples (*FP*), and the number of undetected positive samples (*FN*). Then, we calculated the precision rate and recall rate and finally obtained the value of $F_1$. The results are shown in Table 1.

**Table 1.** Shine-Muscat grape segmentation model evaluation.

| *TP* | *FP* | *FN* | $F_1$/% |
|:---:|:---:|:---:|:---:|
| 348 | 5 | 29 | 95.34 |

The results show that the grape segmentation model trained with Mask R-CNN is effective and reliable. It can be used for the rapid extraction of the RoI in hyperspectral images.

### 3.2. Results and Analysis of SSC Prediction Test

After obtaining the average spectral reflectance of the RoI of 100 grapes in the modeling set, we obtained the reflectance spectra of the modeling set. Below, each curve represents a grape sample in Figure 10.

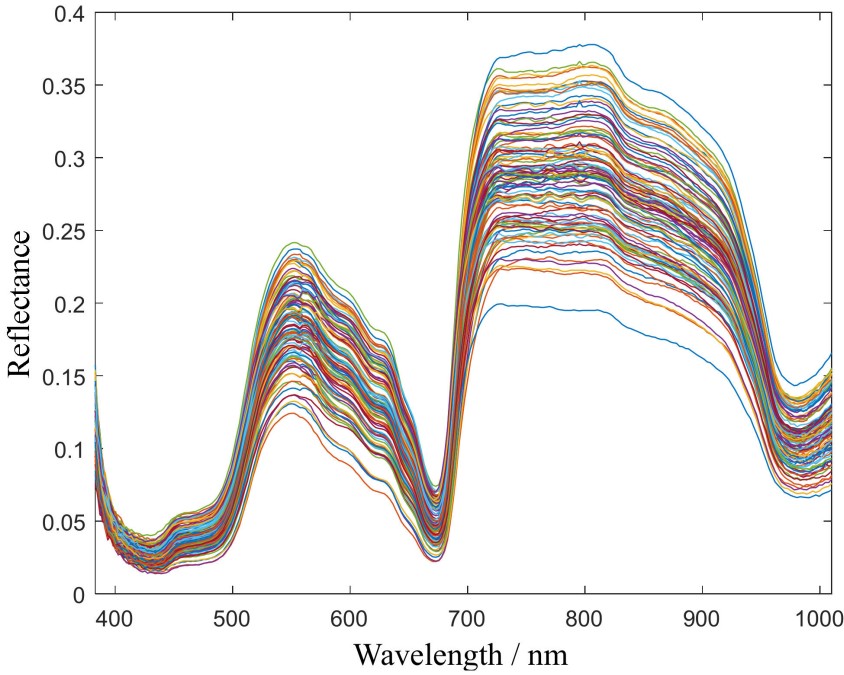

**Figure 10.** Reflectance spectra of modeling set.

### 3.2.1. Eliminating Outlier Samples

In this test, the Mahalanobis distance method was used to eliminate outlier samples, and the Mahalanobis distance and critical threshold of each sample were calculated. Most of the samples were within the critical threshold, and samples exceeding the critical threshold were considered to be outlier samples. Through observation, we could determine that the samples with the serial numbers 54, 64, and 78 were obviously outlier samples; hence, they were eliminated (as shown in Figure 11).

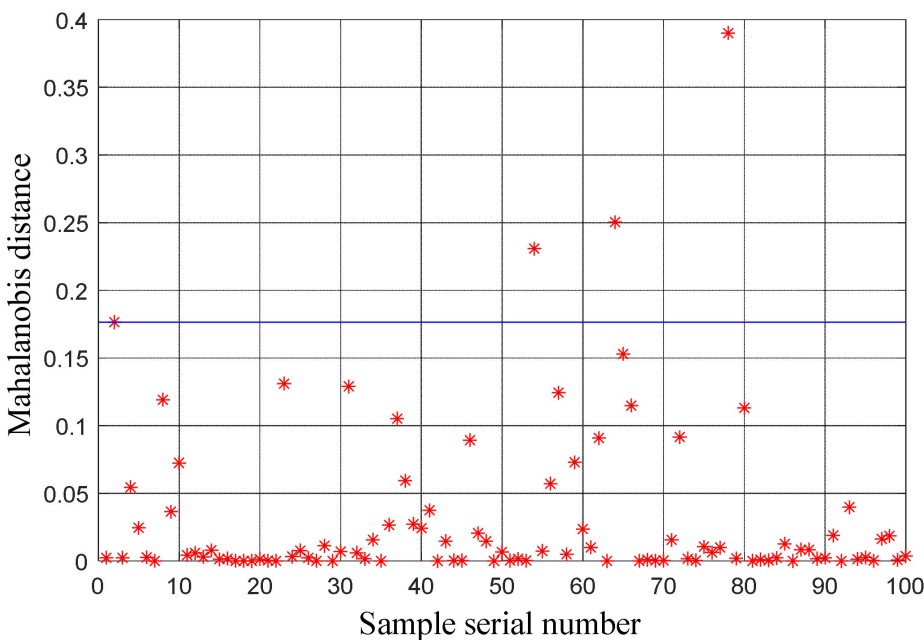

**Figure 11.** Mahalanobis distance between the samples.

3.2.2. Selecting Characteristic Wavelengths

We used different methods in MATLAB R2021b to select feature wavelengths: SPA, MCUVE, SPA-MCUVE (based on the feature wavelengths selected by SPA and then using MCUVE to select feature wavelengths), and MCUVE-SPA. The results are as follows:

1. Used SPA to select characteristic wavelengths and select 32 wavelengths, namely, 393.7, 407.2, 416.2, 420.7, 423.0, 427.5, 429.8, 482.4, 524.3, 557.2, 573.9, 641.2, 690.2, 715.0, 795.5, 798.0, 823.6, 859.6, 875.2, 911.8, 946.1, 951.4, 962.0, 972.6, 986.0, 988.7, 991.4, 999.4, 1002.1, 1004.8, 1007.5, and 1010.2 (unit: nm).
2. Used MCUVE to select characteristic wavelengths and select 35 wavelengths, namely, 391.5, 402.7, 454.8, 468.6, 484.7, 531.3, 536.0, 543.1, 552.5, 557.2, 564.4, 573.9, 617.0, 638.8, 653.4, 680.4, 687.8, 692.7, 707.5, 712.5, 732.4, 742.4, 760.0, 815.9, 841.5, 859.6, 864.8, 875.2, 909.2, 924.9, 943.4, 954.0, 964.7, 986.0, and 1010.2 (unit: nm).
3. Used SPA-MCUVE to select characteristic wavelengths and select 13 wavelengths, namely, 393.7, 423.0, 482.4, 557.2, 573.9, 715.0, 795.5, 859.6, 875.2, 911.8, 962.0, 988.7, and 1010.2 (unit: nm).
4. Used MCUVE-SPA to select characteristic wavelengths and select 15 wavelengths, namely, 391.5, 557.2, 573.9, 638.8, 680.4, 687.8, 712.5, 742.4, 760.0, 859.6, 909.2, 943.4, 964.7, 986.0, and 1010.2 (unit: nm).

In addition, in order to investigate the effects of spectral pretreatment and the selection of feature wavelengths, a control group was set up: no feature selection, and first derivative pretreatment without feature selection.

3.2.3. Establishment and Comparison of SSC Prediction Models

We used PLSR to establish multivariate linear regression models. The measured and predicted values of the modeling set and test set are shown in Figure 12.

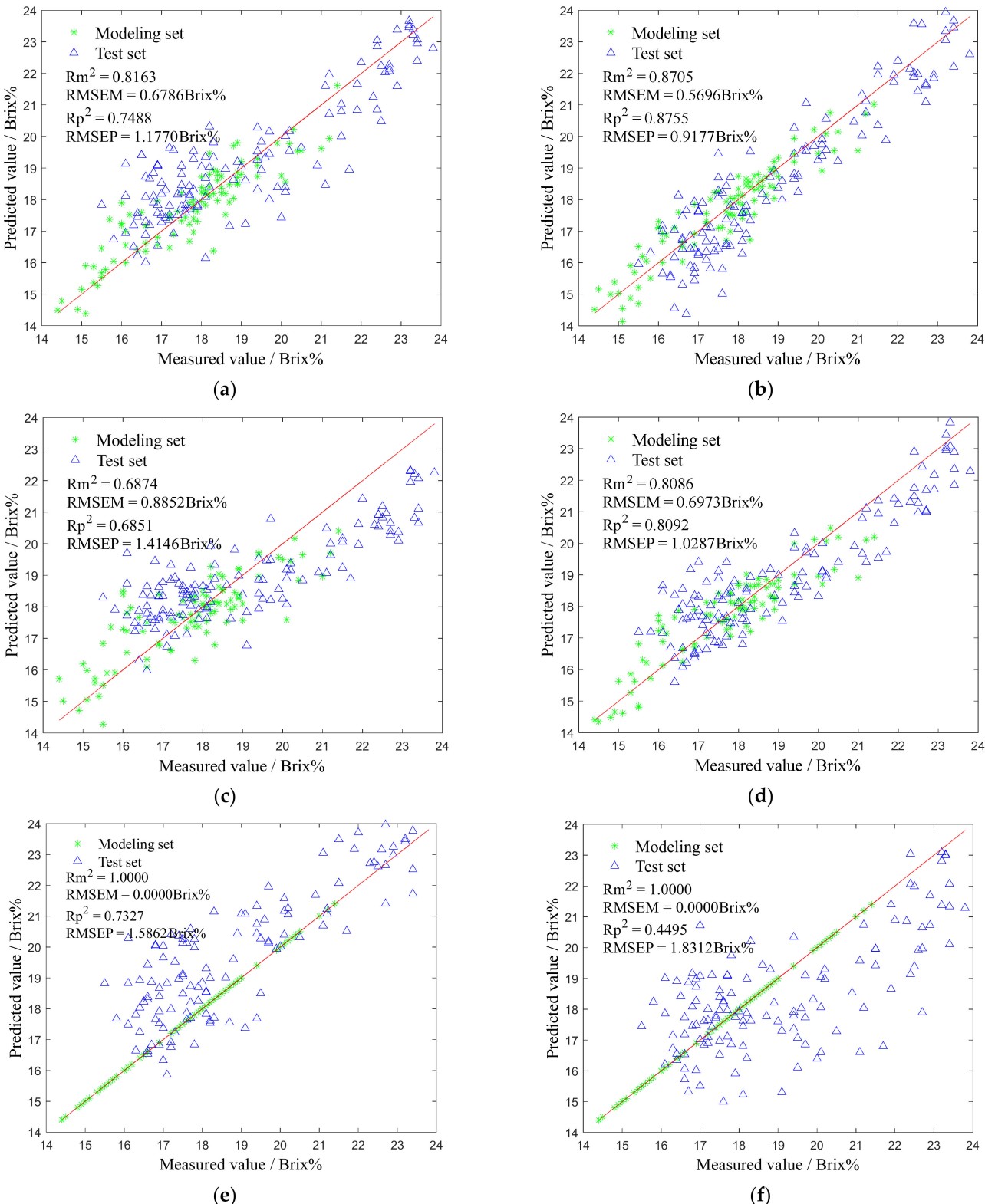

**Figure 12.** The prediction results of the four regression models. (**a**) SPA. (**b**) MCUVE. (**c**) SPA-MCUVE. (**d**) MCUVE-SPA. (**e**) No selection. (**f**) 1-Der without selection.

The model performance indices are shown in Table 2. The main evaluation indicators are $R_p^2$ and RMSEP of the test set, and the auxiliary evaluation indicators are $R_m^2$ and RMSEM of the modeling set. It can be concluded that in terms of accuracy, the ranking

of the models from high to low is as follows: MCUVE, MCUVE-SPA, SPA, no selection, SPA-MCUVE, 1-Der without selection.

**Table 2.** Comparison of different models.

| Method | Characteristic Wavelengths | Modeling Set | | Test Set | |
|---|---|---|---|---|---|
| | | $R_m^2$ | RMSEM (Brix%) | $R_p^2$ | RMSEP (Brix%) |
| SPA | 32 | 0.8163 | 0.6786 | 0.7488 | 1.1770 |
| MCUVE | 35 | 0.8705 | 0.5696 | 0.8755 | 0.9177 |
| SPA-MCUVE | 13 | 0.6874 | 0.8852 | 0.6851 | 1.4146 |
| MCUVE-SPA | 15 | 0.8086 | 0.6973 | 0.8092 | 1.0287 |
| No selection | 256 | 1.0000 | 0.0000 | 0.7327 | 1.5862 |
| 1-Der without selection | 256 | 1.0000 | 0.0000 | 0.4495 | 1.8312 |

It can be seen from Table 2 that among all the models, the model established using MCUVE to select the characteristic wavelengths had the best performance, and the $R_p^2$ of the test set was 0.8755, which showed good versatility and stability. The models without characteristic wavelengths selection had overfitting. The selection of characteristic wavelengths should not be too many or too few. In addition, it can also be seen that using pretreatment methods such as first derivative can lead to a decrease in model performance.

After comprehensive consideration, we chose the model established using 35 wavelengths selected with MCUVE. The equation of the model is shown in Figure A1, where *Y* is the value of the grape SSC (unit: Brix%), and $X_k$ is the standard reflectance of the grape at wavelength *k*.

There has been some research on the selection of characteristic wavelengths for different grape varieties, for example, Red Globe [12,36], Wink [13], and various wine grapes (Cabernet Sauvignon, Nebbiolo, Merlot, Shiraz, etc.) [37–39]. By comparing these studies, it can be generally concluded that the wavelengths related to SSC are as follows: 450–460 nm, 670–680 nm, 720–730 nm, 950–980 nm.

### 3.3. Validation Results of Improved Non-Destructive Detection Method for SSC in Bunch-Harvested Grapes

Two key models, as outlined above, were established: the grape segmentation model based on DL and the SSC prediction model based on HSI. Finally, the validation of the non-destructive detection technology for the SSC of bunch-harvested grapes was carried out, and the sample is shown in Figure 8, including eight bunches of grapes.

When imaging within the platform, we assumed that the reflectance spectrum of the same object at different heights (i.e., distance from the light source) was uniform. In reality, when placing a bunch of grapes in the platform, due to the irregular shape, the distance between different grapes and the light source was slightly different. This could lead to slight deviations in hyperspectral data.

#### 3.3.1. Grape Segmentation and Removal of Overbright Areas

The identification and segmentation results of a single grape, obtained using the Mask R-CNN image instance segmentation model, are shown in Figure 13a,b.

By comparing the grayscale images at each wavelength, it was found that the overbright area was the most evident at the wavelength of 670 nm. Therefore, the grayscale image at the wavelength of 670 nm was binarized and inverted as a mask to remove the overbright area. It was multiplied with the original image to obtain the corresponding pixels of the grapes without overbright areas (as shown in Figure 13c).

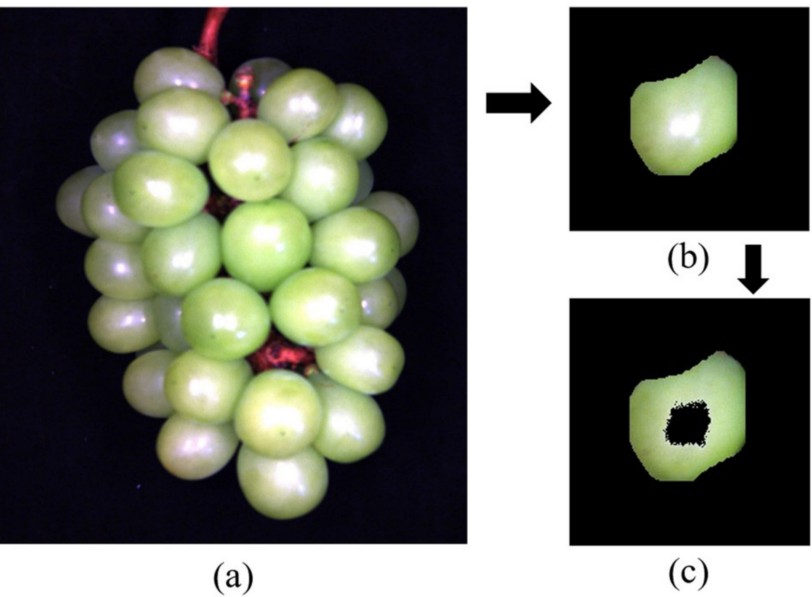

**Figure 13.** Obtaining the pixels of a single grape. (**a**) A bunch of grapes. (**b**) Segmentation of single grape by Mask R-CNN. (**c**) Removing the overbright area.

Through the process outlined above, we could obtain the corresponding pixels of each grape in the bunch.

### 3.3.2. Prediction of SSC for Bunch-Harvested Grapes

After obtaining the corresponding pixels of each grape, the average reflectance spectrum for each grape was calculated. Using the SSC prediction model, the predicted value of the SSC of each grape was obtained.

In grape production, it is necessary to obtain the average, maximum, and minimum values of the SSC of the whole bunch of grapes, as an indicator of the maturity classification, to determine whether it is suitable for picking [40]. The results are shown in Table 3.

**Table 3.** SSC prediction results for bunch-harvested grapes.

| SSC (Brix%) | 1st Bunch | | | 2nd Bunch | | | 3rd Bunch | | | 4th Bunch | | |
|---|---|---|---|---|---|---|---|---|---|---|---|---|
| | Avg | Min | Max | Avg | Min | Max | Avg | Min | Max | Avg | Min | Max |
| Measured value | 16.78 | 15.5 | 18.8 | 17.18 | 15.4 | 18.4 | 15.33 | 14.2 | 16.3 | 16.47 | 14.6 | 18.2 |
| Predicted value | 17.43 | 16.21 | 18.76 | 17.39 | 15.36 | 18.93 | 15.79 | 13.84 | 17.64 | 16.38 | 14.89 | 17.88 |
| Error | 0.65 | 0.72 | 0.04 | 0.21 | 0.04 | 0.53 | 0.46 | 0.36 | 1.34 | 0.09 | 0.29 | 0.22 |
| **SSC (Brix%)** | **5th Bunch** | | | **6th Bunch** | | | **7th Bunch** | | | **8th Bunch** | | |
| | Avg | Min | Max | Avg | Min | Max | Avg | Min | Max | Avg | Min | Max |
| Measured value | 15.57 | 14.2 | 16.7 | 15.94 | 14.8 | 16.9 | 18.68 | 17.0 | 21.6 | 23.29 | 20.9 | 25.5 |
| Predicted value | 16.62 | 15.51 | 17.30 | 16.88 | 16.14 | 17.72 | 19.42 | 17.54 | 20.49 | 23.04 | 21.07 | 24.83 |
| Error | 1.05 | 1.31 | 0.60 | 0.94 | 1.34 | 0.82 | 0.74 | 0.54 | 1.11 | 0.25 | 0.17 | 0.67 |

It can be seen that among the eight bunches of grapes, the prediction error of the average SSC value of the whole bunch was between 0.09 and 1.05 Brix%, the prediction error of the minimum SSC value of the whole bunch was between 0.04 and 1.34 Brix%, and the prediction error of the maximum SSC value of the whole bunch was between 0.04 and 1.34 Brix%. The test results prove that the SSC non-destructive detection method was also effective and accurate for the whole bunch of grapes imaged.

According to the average coordinates of the grape pixel points, the predicted SSC results of each grape were displayed in the color map to obtain the visualization result (as shown in Figure 14).

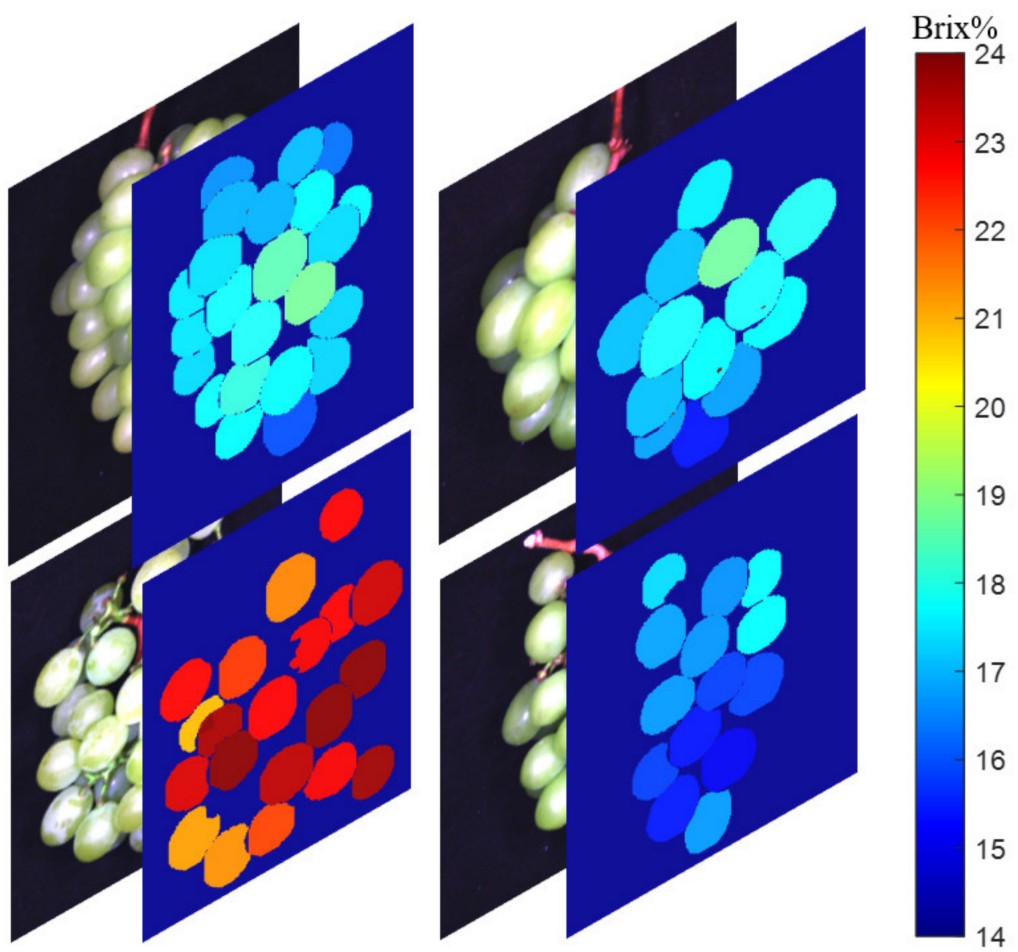

**Figure 14.** Visualization of SSC non-destructive detection results for bunch-harvested grapes.

3.3.3. Efficiency Comparison between Existing Methods and Improved Methods

In recent years, researchers have successively carried out research on the non-destructive testing of grape soluble solids content based on HSI. In 2017, Xu Li et al. established a prediction model for SSC based on near-infrared hyperspectral images (500~1000 nm) for red grapes and achieved a good accuracy [14]. Such existing methods usually use ENVI software (version number from 4.7 to 5.3) to manually select RoIs in order to obtain the average spectral reflectance of grapes.

Therefore, the existing method presented in article [14] and the improved method proposed in this research were used to obtain the reflectance spectra of grapes, and the non-destructive detection of SSC was carried out on eight bunches of grapes. Then, we compared the speed and precision of the two methods on the same computer in a laboratory. The existing methods use ENVI 5.3 for manual RoI selection, as indicated by the colored rectangular blocks in Figure 15a. The improved method uses Mask R-CNN for recognition and segmentation and uses the threshold method to remove overbright areas, as shown in Figure 15b.

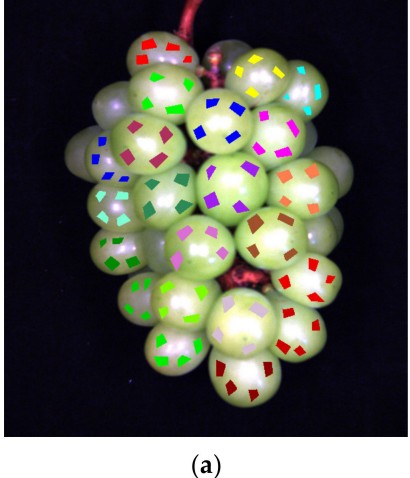
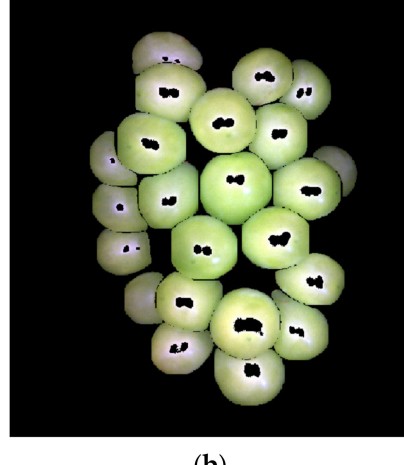

(**a**)                                                (**b**)

**Figure 15.** Method for obtaining grape spectral data. (**a**) Existing method (different colors represent different RoIs). (**b**) Improved method.

The timing started from the moment when the spectral data were extracted based on the hyperspectral image, and the timing stopped when the prediction of the SSC of the eight bunches of grapes was completed. For a total of 157 grapes, the existing method took 35 min and 15 s, and the improved method took 2 min and 8 s. The performance comparison is shown in Table 4.

**Table 4.** Comparison between existing and improved methods.

| Method | Speed (Grapes/Sec) | Model Accuracy | |
| --- | --- | --- | --- |
| | | $R_p$ | RMSEP (Brix%) |
| Article [14] method | 0.074 | 0.9620 | 0.3720 |
| Improved method | 1.227 | 0.9357 | 0.9177 |

It can be seen from Table 4 that although the model accuracy of the improved method proposed in this research was slightly lower than that of the model proposed in [14], the average prediction speed per grape of the improved method was 16.6 times that of the former, which proves that the improved method greatly improves the performance and non-destructive detection efficiency.

## 4. Conclusions

In this research, by establishing two detection models, namely, the Shine-Muscat image segmentation model using Mask R-CNN based on deep learning and the SSC prediction model based on near-infrared hyperspectral images (400~1000 nm), the existing non-destructive detection method of grape SSC was improved. The high-efficiency non-destructive detection of the SSC of bunch-harvested grapes was realized. The main conclusions are as follows:

1.  The $F_1$ score of the Shine-Muscat segmentation model is 95.34%, and it can effectively identify and segment single fruits from bunch-harvested grapes so as to realize automatic and efficient RoI selection using hyperspectral images.
2.  The SSC prediction model is established using 35 characteristic wavelengths, the $R^2$ of the modeling set and the test set reach 0.8705 and 0.8755, respectively, and the RMSE values are 0.5696 Brix% and 0.9177 Brix%, respectively. Accurate prediction of grape SSC based on HSI is realized.
3.  Using eight bunches of grapes for verification, the prediction error of the average SSC value of the whole bunch is between 0.09 and 1.05 Brix%, the prediction error

of the minimum SSC value of the whole bunch is between 0.04 and 1.34 Brix%, and the prediction error of the maximum SSC value of the whole bunch is between 0.04 and 1.34 Brix%. This shows that this method can provide accurate reference data for actual production.

4. Using eight bunches of grapes to compare the existing method and the improved method in terms of efficiency, the average detection speed of the existing method is 0.074 grapes per second, and that of the improved method is 1.227 grapes per second, which is 16.6 times that of the former. The results prove that the improved method can greatly improve the efficiency of non-destructive detection based on HSI.

This method overcomes the shortcomings of the traditional SSC detection method, which requires one to destroy the sample, and improves the existing non-destructive detection method using HSI, saving manpower and material resources. In addition, the visual display of SSC directly reflects the differences in the SSCs of different grape grains within the same bunch, which helps one to grade the grape quality.

However, the study still had the following shortcomings: it is difficult to integrate all functions in a single device; the portability is insufficient; and the degree of automation is low. In order to be applied to production and sales, the method needs to be improved: from an algorithm perspective, it is necessary to integrate the functions of hyperspectral imaging, spectral correction, image segmentation, and SSC prediction into one project (currently, the integration of image segmentation and SSC prediction has been achieved). From a hardware perspective, a hyperspectral imager and computer can be installed on a field inspection robot to achieve unmanned automatic data collection and analysis in the field, and the data analysis results can be uploaded to a cloud server.

**Author Contributions:** Conceptualization, J.Z., Q.H. and H.L.; Methodology, J.Z., Q.H. and H.L.; Validation, Q.H. and Y.X.; Resources, Y.X.; Data curation, Q.H.; Writing—original draft, Q.H.; Writing—re-view & editing, B.L. and S.X.; Visualization, Q.H.; Supervision, B.L. and S.X. All authors have read and agreed to the published version of the manuscript.

**Funding:** This research was funded by Laboratory of Lingnan Modern Agriculture Project (NT2021009), The Project of Collaborative Innovation Center of Guangdong Academy of Agricultural Sciences (XTXM202201), the 2020 Provincial Agricultural Science and Technology Innovation and Extension System Construction Project (2020KJ256) and the President's Foundation of Guangdong Academy of Agricultural Sciences (201940).

**Institutional Review Board Statement:** Not applicable.

**Informed Consent Statement:** Not applicable.

**Data Availability Statement:** Not applicable.

**Conflicts of Interest:** The authors declare no conflict of interest.

## Appendix A

$$\begin{aligned} Y =\ & 14.2358 + 92.3637 X_{391.5} - 68.7432 X_{402.7} + 161.9487 X_{454.8} - 557.9526 X_{468.6} + 337.3863 X_{484.7} \\ & + 37.8253 X_{531.3} - 68.2912 X_{536.0} + 213.5366 X_{543.1} - 561.0685 X_{552.5} - 141.5644 X_{557.2} \\ & + 276.2822 X_{564.4} + 520.2642 X_{573.9} - 671.9642 X_{617.0} + 360.4839 X_{638.8} - 156.2147 X_{653.4} \\ & + 593.0689 X_{680.4} - 474.1977 X_{687.8} + 59.9195 X_{692.7} - 144.9838 X_{707.5} + 492.5638 X_{712.5} \\ & + 673.6929 X_{732.4} - 742.2529 X_{742.4} - 277.0061 X_{760.0} - 79.7755 X_{815.9} - 29.5035 X_{841.5} \\ & + 585.5963 X_{859.6} - 260.0686 X_{864.8} + 128.7665 X_{875.2} - 323.8834 X_{909.2} - 157.4050 X_{924.9} \\ & + 329.8163 X_{943.4} + 101.7265 X_{954.0} + 192.8572 X_{964.7} - 441.4492 X_{986.0} + 85.1801 X_{1010.2} \end{aligned}$$

**Figure A1.** Multivariate linear regression model for SSC based on reflectance spectrum.

**Table A1.** Instance segmentation baselines with Mask R-CNN.

| Name | lr Sched | Train Time (s/iter) | Inference Time (s/im) | Train Mem (GB) | Box AP | Mask AP |
|------|----------|---------------------|----------------------|----------------|--------|---------|
| R50-C4 | 1× | 0.584 | 0.110 | 5.2 | 36.8 | 32.2 |
| R50-DC5 | 1× | 0.471 | 0.076 | 6.5 | 38.3 | 34.2 |
| R50-FPN | 1× | 0.261 | 0.043 | 3.4 | 38.6 | 35.2 |
| R50-C4 | 3× | 0.575 | 0.111 | 5.2 | 39.8 | 34.4 |
| R50-DC5 | 3× | 0.470 | 0.076 | 6.5 | 40.0 | 35.9 |
| R50-FPN | 3× | 0.261 | 0.043 | 3.4 | 41.0 | 37.2 |
| R101-C4 | 3× | 0.652 | 0.145 | 6.3 | 42.6 | 36.7 |
| R101-DC5 | 3× | 0.545 | 0.092 | 7.6 | 41.9 | 37.3 |
| R101-FPN | 3× | 0.340 | 0.056 | 4.6 | 42.9 | 38.6 |
| X101-FPN | 3× | 0.690 | 0.103 | 7.2 | 44.3 | 39.5 |

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
