# Peer review of "Research on an Improved Non-Destructive Detection Method for the Soluble Solids Content in Bunch-Harvested Grapes Based on Deep Learning and Hyperspectral Imaging"

_applsci, doi:10.3390/app13116776_

Round 1
Reviewer 1 Report
The present study focuses on predicting the Brix content of a single grape cultivar using a hyperspectral camera. The main novelty in this work lies in the usage of segmentation method to identify the bunches grapes, where most studies employing hyperspectral imagery draw the regions of interest by hand or perform pixel-based classification; it doesn't introduce any novel algorithms in terms of processing but rather uses existing solutions. Thus the novelty is on the low side. The paper is mostly well-written, and the authors have done a commendable effort to describe all the steps in their process. However, there is a lack of comparison of e.g. between two different methods to perform segmentation, or between methods to predict the SSC at the second stage. This is another shortcoming of the paper. There is also a lack of discussion on the selected wavelengths that are used at the second stage. Please find my more concrete remarks below:
1. The introduction does not sufficiently mention past studies that have i) used point spectroscopy to establish that one can indeed link sugar content to hyperspectral signatures, e.g. doi: 10.1016/j.foodchem.2022.134321, 10.3390/s23031065, ii) other studies that have used HSI even though they defined the RoI manually. Particularly for ii) the authors extremely briefly gloss over 5 papers (line 59) not doing them any justice. I could argue (even though I don't hold that opinion!) that if you just apply a pre-trained DL segmentation model (e.g., yolo, mask-CNN) you can extract the grapes and then by applying the mechanisms of the HSI works you didn't expand on, the problem has already been solved and the novelty of the present work is low; thus these papers that used the HSI to predict the sugar content are the most crucial in this approach.
2. Line 92 to 94: Please define what you mean by "20% mature" etc.
3. The sampling schema seems odd, could you please tell us to what phenological growth stages of the grapevine (e.g. BCCH scale) these correspond to?
4. Figure 3 shows grape berries but Figure 4 shows bunches. Please address this confusion as one thinks that these are the training data.
5. Line 95, please mention the model of the camera if it has a name (I couldn't find it via google). Please mention if this is a snapshot camera or push-broom camera. Please mention explicitly the protocol used to collect the hyperspectral images. Please also include the spectral resolution.
6. Line 97: 'is a halogen lamp' => I see 4 in Figure 3?
7. Line 126: 'approximately 50 images' you had 23 bunches, how did you get 50 images? Is one image comprised of multiple bunches as hinted by in Figure 4? (I think not). Did you rotate some of the bunches?
8. Lines 132-135: This is critical. Does this mean that the model cannot identify grapes with slight damage or if they have high occlusion?
9. Section 2.2.3: Why not use IoU?
10. Line 230: Was there enough juice even in low SSC to replicate the measurement three times?
11. Lines 271 to 276: I am missing why you didn't compare in the paper PLSR vs feature selection + MLR, or even other more elaborate regression techniques. Particularly considering that one uses a 'heavy-duty' DL model to perform the segmentation task, the computational cost is not an issue here.
12. Section 3.1.3: Could you please share some examples of FP? This seems interesting!
13. Section 3.2.1: Why did these outliers occur? Are they completely erronneous measurements or are they actually data (e.g., slightly damaged berries)?
14. I am missing an interpetation of the bands selected in section 3.2.2 and a comparison with other works.
15. Line 387: "We used PLSR to establish MLR models ..." is this correct?
16. Lines 405 to 406: Still, I think the authors should test different pre-treatments (e.g. first-derivative) to see if their results could be slightly improved.
17. Figure 14 could be improved by using color to indicate the Brix content.
18. Table 5 and the comparison between speeds is in my humble opinion not necessary. 1) Did [14] and the present use the same hardware? You used 2xGPUs if I saw correctly. 2) The speed in this offline process where one first harvest the grapes and then takes them to lab etc. is not a hindering factor.
19. Line 502: stud => study
20. Table 2 and generally, I would also have included the RPIQ metric.
Overall the manuscript is fine, some minor issues are noted above.
Reviewer 2 Report
Your manuscript presents an interesting and valuable study on improving the non-destructive detection of grape SSC using a deep learning-based segmentation model and a hyperspectral imaging-based prediction model. The research is well-structured, and the results are clearly presented. However, there are a few aspects that could benefit from revisions and clarifications. Please consider the following comments and suggestions:
In the introduction, consider providing more context on the importance of grape SSC detection in the agriculture and food industries to help readers better understand the significance of your research.
The Shine-Muscat segmentation model and the SSC prediction model are well-described. However, it would be helpful if you could provide a brief explanation of the rationale behind choosing the Mask R-CNN model for segmentation and the specific model used for the SSC prediction.
In the results section, consider discussing any limitations or potential biases that may have influenced the performance of the models, as well as any assumptions made during the research.
In the conclusion, you mention some shortcomings of the study, such as the difficulty of integrating all functions into a single device, insufficient portability, and low degree of automation. It would be helpful if you could briefly elaborate on these issues and suggest possible future research directions to address them.
Overall, your manuscript makes a valuable contribution to the field, and addressing these comments will further enhance the quality and impact of your research.
Minor editing of the English language is required throughout the manuscript to improve clarity and readability. Ensure that the text is free from grammatical errors, awkward phrasing, and inconsistencies in terminology.
Author Response
Point 1: In the introduction, consider providing more context on the importance of grape SSC detection in the agriculture and food industries to help readers better understand the significance of your research.
Response 1: Thanks for your advice. We will provide more context on the importance of grape SSC detection in food industries, as well as the limitations of current non-destructive detection methods.
Point 2: The Shine-Muscat segmentation model and the SSC prediction model are well-described. However, it would be helpful if you could provide a brief explanation of the rationale behind choosing the Mask R-CNN model for segmentation and the specific model used for the SSC prediction.
Response 2: Thanks for your advice. The reason for choosing Mask R-CNN is because it combines object detection and semantic segmentation, which can not only accurately segment the classification of "grape berry", but also distinguish different grape berries. The model used for the SSC prediction is a multivariate linear regression model, which have one dependent variable and multiple independent variables. It’s shown in Figure A1 in Appendix A.
Point 3: In the results section, consider discussing any limitations or potential biases that may have influenced the performance of the models, as well as any assumptions made during the research.
Response 3: Thanks for your advice. When imaging within a hyperspectral imaging platform, we assumed that the reflectance spectrum at different heights (i.e. distance from the light source) is uniform. However, in reality, when placing a bunch of grapes in the platform, due to the irregular shape, the distance between different grapes and the light source is not the same. This can lead to slight deviations in hyperspectral data.
Point 4: In the conclusion, you mention some shortcomings of the study, such as the difficulty of integrating all functions into a single device, insufficient portability, and low degree of automation. It would be helpful if you could briefly elaborate on these issues and suggest possible future research directions to address them.
Response 4: Thanks for your advice. Specifically, from a software perspective, it is necessary to integrate the functions of hyperspectral imaging, spectral correction, image segmentation, and SSC prediction into one project (currently, the integration of image segmentation and SSC prediction has been achieved). From a hardware perspective, a hyperspectral imager and computer can be installed on a field inspection robot to achieve unmanned automatic data collection and analysis in the field, and the data analysis results can be uploaded to a cloud server.
Reviewer 3 Report
1. Brief summary
In their study, the authors by creating two detection models, namely Shine-Muscat im-473 age segmentation model using Mask R-CNN based on deep learning and SSC-474 prediction model based on near-infrared (400–1000 nm) hyperspectral imaging. . ), improved the existing non-destructive method for the detection of (dry soluble matter) SSC of grapes in whole bunches.
The method is based on 68 near-infrared (400~1000nm) hyperspectral images. Taking the variety Shine-Muscat, whose maturity is difficult to distinguish with the naked eye, as the object of study, and the Mask R-CNN image segmentation algorithm based on the DL-71 image segmentation thinking model and SSC prediction, a model was created to improve the efficiency of non-destructive control of grape maturity.
At the same time, the authors successfully used the Mask R-CNN image instance segmentation algorithm [17], based on deep learning.
In the production of grapes, it is necessary to obtain the average, maximum and minimum SSC values of a whole bunch of grapes as an indicator of the degree of ripeness to justify the timing of harvesting. The authors achieved an error in the prediction of the average SSC for the entire bunch of grapes from 0.09 to 1.05% Brix, errors in the prediction of the minimum SSC for the whole bunch are from 0.04 to 1.34% Brix, and errors in the prediction of the maximum the SSC values of the entire group range from 0.04 to 1.35% Brix. The test results prove that the SSC non-destructive testing method is also effective and accurate for the whole bunch of grapes.
2. Comments on the general concept of the article
Undoubtedly, the authors, with their work, make a new contribution to the modern methodology for using hyperspectral data to assess the quantitative and qualitative indicators of agricultural products and, in particular, the maturity of grapes, by the content of dry soluble matter. The authors develop a methodology for combining the method of parametric use of hyperspectral data and the method of image segmentation using a deep learning algorithm.
A significant drawback of the work is the lack of justification for using such a complex implementation algorithm and the technical means used to solve a fairly simple problem of quantifying the dry matter content in grapes. The authors did not reflect the possibilities of a purely parametric assessment approach using only hyperspectral data and their information content in relation to the content of soluble dry matter.
The authors themselves indicate in their conclusions that it is difficult to integrate all the functions in one device, there is a lack of portability and a low degree of automation. More refinement is required so that the method of non-destructive determination of the internal quality of grapes can be applied to production and sale.
3 general recommendations
Given the high scientific and technical level of the study, the proposed article can be recommended for publication.
Author Response
Point 1: A significant drawback of the work is the lack of justification for using such a complex implementation algorithm and the technical means used to solve a fairly simple problem of quantifying the dry matter content in grapes. The authors did not reflect the possibilities of a purely parametric assessment approach using only hyperspectral data and their information content in relation to the content of soluble dry matter.
Response 1: Sorry for not emphasizing it clearly. As shown in Table 5, the method proposed in this article has been compared in efficiency with Ref [14]. Ref [14] only used hyperspectral imaging technology to predict the SSC of grapes. It can be seen that due to the combination of deep learning in this article, the non-destructive detection efficiency has been greatly improved.
Point 2: The authors themselves indicate in their conclusions that it is difficult to integrate all the functions in one device, there is a lack of portability and a low degree of automation. More refinement is required so that the method of non-destructive determination of the internal quality of grapes can be applied to production and sale.
Response 2: Thanks for your advice. Specifically, from a software perspective, it is necessary to integrate the functions of hyperspectral imaging, spectral correction, image segmentation, and SSC prediction into one project (currently, the integration of image segmentation and SSC prediction has been achieved). From a hardware perspective, a hyperspectral imager and computer can be installed on a field inspection robot to achieve unmanned automatic data collection and analysis in the field, and the data analysis results can be uploaded to a cloud server.
Reviewer 4 Report
This manuscript discusses an improved method to determine the ripeness of grapes through an indirect measure of the Soluble Solids Content. The approach promises an improved, non-destructive method. Certainly, the subject matter is appropriate for the journal. The general layout of the paper seems appropriate as does the referencing. The language is mostly passable but there are some places where it would really help to fix it. In general, I like the topic and think it is a good paper to publish as it provides, it appears, a relatively simple and useful hyperspectral technique.
My concern here is that the paper seems to me to be unclear (mostly from being incomplete) in a number of places. There are a lot of terms used that I don’t think would be understood by the broad readership of the journal. I would like to point out some of the areas that I find objectionable. I believe that the significant rewrite is going to be needed.
Some of the issues are lack of explanation. For example, the Brix% is never explained. The digital refractometer, which provides the “true” value of the SSC is not really explained – how it works would seem relevant here. You don’t use the term ground-truth or “true” value, which would make things clearer. HSIA is not written out – presumably it is a company name (is it Hesai Group?)
On the measurements, there are also some explanation issues. What is the “white” board? Is it spectralon or some other material. I think that more detail here is warranted. The HSI system is a pushbroom and the camera is mechanically scanned across the grapes? What is the spectral resolution?
On line 220 you speak about saturated areas? Is it really saturated? Elsewhere you refer to “overbright” areas (lines 293, for example.) Why is the measurement allowed to saturate? In many instruments having saturated pixels anywhere in the frame negatively impacts the quality of the data (it can drive the dark level high or low.) And in a controlled situation like this it isn’t usually necessary. Perhaps, it is not a saturation problem but instead is a specular reflection problem that results in a contrast issue? And if that is the case, is the real problem that the overbright areas are only sampling the skin and not the interior of the grape?
The use of the Mahalanobis distance needs more explanation. It is the distance between a point and a distribution. What is the distribution? I think that I know but you need to say it explicitly. And how are you calculating it? Is there an explanation for the outliers?
Line 297 “Since the light source used for imaging in the dark box comes from a halogen lamp, there is a problem in that some areas are too bright…” This suggests that the overbright areas occur because the light is a halogen. The logic flow there is difficult for me to understand.
On line 467 you are comparing Article 14 method with your own. There are two things happening here – the selection of the position of the measured spectra that are used and the spectra themselves. In Ref 14, they only did the second part of that, it appears. I think that a more complete comparison makes some sense. I cannot see the full ref 14 paper but the abstract doesn’t seem to suggest that there was automatic position description but does say that there were several corrections made to the spectra. Di you do those here?
I can understand the need to use AI/ML for the shapes but it isn’t obvious why the spectral part of this problem needs it. Is there really no physical change in the spectra that follows the SSC?
In any event, I hope that you are willing to rewrite some portions of the paper. I think it could be quite good.
It isn't so much that the english is poor - the paper mostly needs additional explanation. Yes, some english could be improved but it isn't the main problem.
Round 2
Reviewer 1 Report
The authors have done a commendable effort to improve their manuscript and have addressed sufficiently the majority of my remarks. Still a few points needs to be addressed.
1. In my opinion, the authors still need to improve the introduction section and even refer to studies that employ point spectroscopy; after all, it's on point spectra that they are operating to determine SSC after extracting them from the HSI, so past works should be cited.
2. Line 142: citation for LabelMe?
3. Regarding the model, I still believe the paper would improve if you included PLSR (no feature selection) VS PLSR (with feature selection) to substantiate the claim that you have enhanced accuracy results. I would also include (even in just a sentence) that you tested the first derivative spectra and saw diminished results. These are critical in my opinion for future researchers.
4. The paper would also improve if you include a discussion about the selected wavelengths and provide an interpretation thereof or a comparison with other studies (in e.g. white wine grape varieties) to see if the absorption bands the study picks are the ones that are associated with sugars. It's not necessary for it to be the Sine Muscat to provide a comparison.
5. Section 5 / Line 524 should be deleted? Or renamed CRediT?
